# BiTAT: Neural Network Binarization with Task-dependent Aggregated Transformation

## Abstract

Neural network quantization aims to transform high-precision weights and activations of a given neural network into low-precision weights/activations for reduced memory usage and computation while preserving the performance of the original model. However, 1-bit weight/1-bit activations of compactly-designed backbone architectures often used for edge-device deployments result in severe performance degeneration. This paper proposes a novel Quantization-Aware Training method that can effectively alleviate performance degeneration even with extreme quantization by focusing on the inter-weight dependencies, between the weights within each layer and across consecutive layers. To minimize the quantization impact of each weight on others, we perform an orthonormal transformation of the weights at each layer by training an input-dependent correlation matrix and importance vector, such that each weight is disentangled from the others. Then, we quantize the weights based on their importance to minimize the loss of the information from the original weights/activations. We further perform progressive layer-wise quantization from the bottom layer to the top, so that quantization at each layer reflects the quantized distributions of weights and activations at previous layers. We validate the effectiveness of our method on various benchmark datasets against strong neural quantization baselines, demonstrating that it alleviates the performance degeneration on ImageNet and successfully preserves the full-precision model performance on CIFAR-100 with compact backbone networks.

## 1 Introduction

Over the past decade, deep Neural Networks (NN) have achieved tremendous success in solving various real-world problems (Creswell et al., 2018; Gidaris et al., 2018; Chen et al., 2020; Karras et al., 2021; Radford et al., 2021). Recently, network architectures are becoming increasingly larger based on the empirical observations of their improved performance. However, it is increasingly difficult to deploy them on edge devices with limited memory and computational power. Therefore, many recent works focus on building resource-efficient networks to bridge the gap between their scale and actual permissible computational complexity/memory bounds for on-device model deployments. Several works consider designing computation- and memory-efficient architecture modules, while others focus on compressing a given neural network by either pruning its weights (Yoon & Hwang, 2017; He et al., 2020b; Lin et al., 2020a) or reducing the bits used to represent the weights and activations (Bulat et al., 2021; Dbouk et al., 2020; Li et al., 2021). The latter approach, *neural network quantization*, is beneficial for building on-device AI systems since the edge devices oftentimes only support low bitwidth-precision parameters and/or operations. However, it inevitably suffers from the non-negligible forgetting of the encoded information from the full-precision models. Such loss of information becomes worse with extreme quantization into binary neural networks with 1-bit weights and 1-bit activations (Bulat et al., 2021; Zhuang et al., 2019; Qin et al., 2020b).

How can we then effectively preserve the original model performance even with extremely low-precision networks? To address this question, we focus on the somewhat overlooked properties of NN for quantization: the weights in a layer are highly correlated with each other and weights in consecutive layers. Quantizing the weights will inevitably affect the weights within the same layer since they together comprise a transformation represented by the layer. Thus, quantizing the weights and activations at a specific layer will adjust the correlation and relative importance between them. Moreover, it will also largely impact the next layer that directly uses the output of the layer, which together comprise a function represented by the neural network.

| METHOD | BRECQ Li et al. (2021) | DBQ Dbouk et al. (2020) | ReActNet Liu et al. (2020) | Ours |
|---|---|---|---|---|
| $\text{BIT}_w$ / $\text{BIT}_a$ | 2/4 | 4/8 | 1/1 | 1/1 |
| CORRELATION | block | N/A | N/A | block |
| TASK-BASED $Q$ | ✓ | × | × | ✓ |
| STRUCTURED | × | × | × | dynamic |
| APPROACH | PTQ[1] | QAT[2] | QAT | QAT |
| T1@IMGNET | 66.60% | 70.5% | 68.26% | **68.51%** |
| FLOPs $\times 10^7$ | 3.31 | 3.60 | **1.2** | **1.2** |

[1] Post-training Quantization
[2] Quantization-aware Training

Figure 1: **Left: An Illustration of our proposed method.** Weight elements in a layer is highly correlated to each other along with the weights in other layers. Our BiTAT sequentially obtains quantized weights of each layer based on the importance of disentangled weights to others using a trainable orthonormal rotation matrix and importance vector. **Right:** Categorization of relevant and strong quantization methods to ours.

Despite their impact on NN quantization, such inter-weight dependencies have been relatively overlooked. As shown in Figure 1 Right, although BRECQ (Li et al., 2021) addresses the problem by considering the dependency between filters in each block, it is limited to the Post-Training Quantization (PTQ) problem, which suffers from inevitable information loss, resulting in inferior performance. Most recent Quantization-Aware Training (QAT) methods (Dbouk et al., 2020; Liu et al., 2020) are concerned with obtaining quantized weights by minimizing quantization losses with parameterized activation functions, disregarding cross-layer weight dependencies. To the best of our knowledge, no prior work explicitly considers dependencies among the weights for QAT.

To tackle this challenging problem, we propose a new QAT method, referred to as Neural Network **Bi**narization with **T**ask-dependent **A**ggregated **T**ransformation (**BiTAT**), as illustrated in Figure 1 Left. Our method sequentially quantizes the weights at each layer of a pre-trained model based on chunk-wise input-dependent weight importance by training orthonormal dependency matrices and scaling vectors. After quantizing each layer, we fine-tune the subsequent full-precision layers, which utilize the quantized layer as an input for a few epochs while keeping the quantized weights frozen. we aggregate redundant input dimensions for transformation matrices and scaling vectors, significantly reducing the computational cost of the quantization process. Such consideration of inter-weight dependencies allows our BiTAT algorithm to better preserve the information from a given high-precision network, allowing it to achieve comparable performance to the original full-precision network even with extreme quantization, such as binarization of both weights and activations. The main contributions of the paper can be summarized as follows:

- We demonstrate that weight dependencies within each layer and across layers play an essential role in preserving the model performance during quantized training.

- We propose an input-dependent quantization-aware training method that binarizes neural networks. We disentangle the correlation in the weights from across multiple layers by training rotation matrices and importance vectors, which guides the quantization process to consider the disentangled weights' importance.

- We empirically validate our method on several benchmark datasets against state-of-the-art NN quantization methods, showing that it significantly outperforms baselines with the compact neural network architecture.

## 2 RELATED WORK

**Minimizing the quantization error.** Quantization methods for deep neural networks can be broadly categorized into several strategies (Qin et al., 2020a). We first introduce the methods that aim to minimize the weight/activation discrepancy between quantized models and their high-precision counterparts. XNOR-Net (Rastegari et al., 2016) aims to minimize the least-squares error between quantized and full-precision weights for each output channel in layers. DBQ (Dbouk et al., 2020) and QIL (Jung et al., 2019) perform layerwise quantization with parametric scale or transformation functions optimized to the task. Yet, they quantize full-precision weight elements regardless of the correlation between other weights. While TSQ (Wang et al., 2018) and Real-to-Bin (Martinez et al., 2020) propose to minimize the $\ell_2$ distance between the quantized activations and the real-valued network's activations by leveraging intra-layer weight dependency, they do not consider cross-layer dependencies. ProxyBNN (He et al., 2020a) adopts the orthogonal matrix to preserve the

correlation between coordinates while minimizing the quantization error. Recently, BRECQ (Li et al., 2021) and the work in a similar vein on post-training quantization (Nagel et al., 2020) consider the interdependencies between the weights and the activations by using a Taylor series-based approach. However, calculating the Hessian matrix for a large neural network is often intractable, and thus they resort to strong assumptions such as small block-diagonality of the Hessian matrix to make them feasible. BiTAT solves this problem by training the dependency matrices alongside the quantized weights while grouping similar weights together to reduce the computational cost.

**Modifying the task loss function.** BNN-DL (Ding et al., 2019) adds a distributional loss enforcing the weight distributions to be quantization-friendly. Apprentice (Mishra & Marr, 2018) uses knowledge distillation to preserve the knowledge of the full-precision teacher network in the quantized network. However, such methods only put a constraint on the distributional properties of the weights, not the dependencies and the values of the weight elements. CI-BCNN (Wang et al., 2019) parameterizes bitcount operations by exploring the interaction between output channels using reinforcement learning and quantizes the floating-point accumulation in convolution operations based on them. However, reinforcement learning is expensive, and it still does not consider cross-layer dependencies. RBNN (Lin et al., 2020b) achieves a significantly higher cosine similarity between the full-precision weight and its binarization by constraining the model to preserve fewer angular biases.

**Reducing the gradient error.** Liu et al. (2018) devises a better gradient estimator for the sign function used to binarize the activations and a magnitude-aware gradient correction method. PCNN (Gu et al., 2019) proposes a new discrete backpropagation method via projection, where the layerwise trainable function effectively projects the weights at each layer to multiple quantized weights. Re-ActNet (Liu et al., 2020) achieves state-of-the-art performance for binary neural networks by training a generalized activation function for compact network architecture used in Liu et al. (2018). However, their quantizer functions conduct element-wise unstructured compression without considering the change in other correlated weights throughout quantization training. This makes the search process converge to suboptimal solutions since task loss is the only guide for finding the optimal quantized weights, which is often insufficient for high-dimensional and complex architectures. On the other hand, we can obtain a better-informed guide that compels the training procedure to spend more time searching in areas that are more likely to contain high-performing quantized weights.

## 3 WEIGHT IMPORTANCE FOR QUANTIZATION-AWARE TRAINING

We aim to quantize a full-precision neural network into a binary neural network (BNN), where the obtained quantized network is composed of binarized 1-bit weights and activations, which preserves the performance of the original full-precision model. Let $f(\cdot; \mathcal{W})$ be a $L$-layered neural network parameterized by a set of pre-trained weights $\mathcal{W} = \{\boldsymbol{w}^{(1)}, \ldots, \boldsymbol{w}^{(L)}\}$, where $\boldsymbol{w}^{(l)} \in \mathbb{R}^{d_{l-1} \times d_l}$ is the weight at layer $l$ and $d_0$ is the dimensionality of the input. Given a training dataset $\mathcal{X}$ and corresponding labels $\mathcal{Y}$, existing QAT methods Rastegari et al. (2016); Dbouk et al. (2020); Jung et al. (2019); Bethge et al. (2020); Yamamoto (2021); Park & Yoo (2020) search for optimal quantized weights by solving for the optimization problem that can be generally described as follows:

$$\underset{\mathcal{W}, \boldsymbol{\phi}}{\text{minimize}} \ \mathcal{L}_{task}\left(f\left(\mathcal{X}; Q\left(\mathcal{W}; \boldsymbol{\phi}\right)\right), \mathcal{Y}\right), \tag{1}$$

where $\mathcal{L}_{task}$ is a standard task loss function, such as cross-entropy loss, and $Q(\cdot; \boldsymbol{\phi})$ is the weight quantization function parameterized by $\boldsymbol{\phi}$ which transforms a real-valued vector to a discrete, binary vector. Existing works quantize typically by rounding each element of the weights or activation to the nearest quantization value. This is equivalent to minimizing loss terms based on the Mean Squared Error (MSE) between the full-precision weights and the quantized weights at each layer:

$$Q(\boldsymbol{w}) := \alpha^* \boldsymbol{b}^*, \quad \text{where } \alpha^*, \boldsymbol{b}^* = \underset{\alpha \in \mathbb{R}, \boldsymbol{b} \in \{-1, 1\}^m}{\arg\min} \|\boldsymbol{w} - \alpha \boldsymbol{b}\|_2^2, \tag{2}$$

where $m$ is the dimensionality of the target weight. For inference, $\boldsymbol{w}_q = Q(\boldsymbol{w})$ is used. Using this quantizer, QAT methods iteratively search for the quantized weights based on the task loss using stochastic gradient descent-based methods, and the model parameters converge into the ball-like region around the full-precision weights $\boldsymbol{w}$. However, the region around the optimal full-precision weights may contain suboptimal solutions with high errors. We demonstrate such inefficiency of the existing quantizer formulation through a simple experiment in Figure 2. Suppose we have three input points, $\boldsymbol{x}_1, \boldsymbol{x}_2$, and $\boldsymbol{x}_3$, and full-precision weights $\boldsymbol{w}$.

Quantized training of the weight using Equation 2 successfully reduces MSE between the quantized weight and the full-precision, but the task prediction loss using $\boldsymbol{w}_q$ is nonetheless very high.

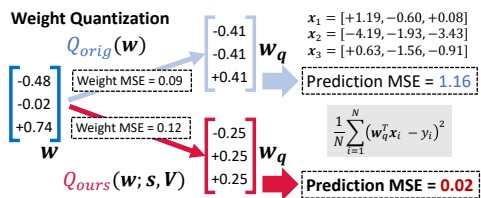

The main source of error comes from the independent application of the quantization process to each weight element: Neural network weights are not independent, but highly correlated, so holding the loss value constant, quantizing (perturbing) one weight will affect the others. Moreover, after quantization, the weight importances can also change significantly.

Figure 2: A simple experiment that cross-layer weight correlation is critical to find well-performing quantized weights during QAT.

Both factors lead to high errors in the pre-activations. On the other hand, our proposed QAT method *BiTAT*, described in Section 4, achieves a quantized model with smaller error. This results from the consideration of the inter-weight dependencies, which we describe in the next subsection.

### 3.1 DISENTANGLING WEIGHT DEPENDENCIES VIA INPUT-DEPENDENT ORTHORNORMAL TRANSFORMATION

How can we, then, find the low-precision subspace, which contains the best-performing quantized weights on the task, by exploiting the inter-weight dependencies? The properties of the input distribution give us some insights into this question. Let us consider a task composed of $N$ centered training samples $\{\boldsymbol{x}_1, \ldots, \boldsymbol{x}_N\} = \mathcal{X} \in \mathbb{R}^{N \times d_0}$. We can obtain principal components of the training samples $\boldsymbol{v}_1, \ldots, \boldsymbol{v}_{d_0} \in \mathbb{R}^{d_0}$ and the corresponding coefficients $\lambda_1, \ldots, \lambda_{d_0} \geq 0$, in descending order. When we optimize a single-layered neural network parameterized by $\boldsymbol{w}^{(1)}$, neurons corresponding to the columns of $\boldsymbol{w}^{(1)}$ are oriented in a similar direction to the principal components with higher variances (i.e., $\boldsymbol{v}_i$ than $\boldsymbol{v}_j$, where $i < j$) that is much more likely to get activated than the others. We apply a change of basis to the column space of the weight matrix $\boldsymbol{w}^{(1)}$ with the bases $(\boldsymbol{v}_1, \ldots, \boldsymbol{v}_{d_0})$:

$$\boldsymbol{V}^{(0)} \widetilde{\boldsymbol{w}}^{(1)} = \boldsymbol{w}^{(1)} \tag{3}$$

$$\widetilde{\boldsymbol{w}}^{(1)} = \boldsymbol{V}^{(0)^\top} \boldsymbol{w}^{(1)}, \tag{4}$$

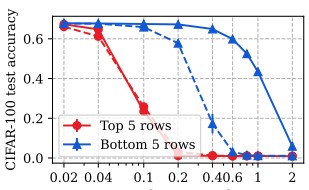

Figure 3: **Solid lines:** Test accuracy of a MobileNetV2 model on CIFAR-100 dataset, after adding Gaussian noise to the top 5 rows and the bottom 5 rows of $\widetilde{\boldsymbol{w}}^{(l)}$ for all layers, considering the dependency on the lower layers. **Dashed lines:** Not considering the dependency on the lower layers. The x axis is in log scale.

where $\boldsymbol{V}^{(0)} = [\boldsymbol{v}_1 \mid \cdots \mid \boldsymbol{v}_{d_0}] \in \mathbb{R}^{d_0 \times d_0}$ is an orthonormal matrix. The top rows of the transformed weight matrix $\widetilde{\boldsymbol{w}}^{(1)}$ will contain more important weights, whereas the bottom rows will contain less important ones. Therefore, the accuracy of the model will be more affected by the perturbations of the weights at top rows than ones at the bottom rows. Note that this transformation can also be applied to the convolutional layer by "unfolding" the input image or feature map into a set of patches, which enables us to convert the convolutional weights into a matrix (The detailed descriptions of the orthonormal transformations for convolutional layers is provided in the supplementary file).

We can also easily generalize the method to multi-layer neural networks, by taking the inputs for the $l$-th layer as the "training set", assuming that all of the previous layer's weights are fixed, as follows:

$$\left\{ \boldsymbol{x}_i^{(l)} = \delta \left( \boldsymbol{w}^{(l)^\top} \boldsymbol{x}_i^{(l-1)} \right) \right\}_{i=1}^{N}, \tag{5}$$

where $\delta(\cdot)$ is the nonlinear transformation defined by both the non-linear activations and any layers other than linear transformation with the weights, such as pooling or Batch Normalization. Then, we obtain the change-of-basis matrix $V^{(l)}$ for layer $l$ by using PCA on $\boldsymbol{x}_i^{(l-1)}$. The impact of transforming the weights is shown in Figure 3. We compute the principal components of each layer in the initial pre-trained model and measure the test accuracy after adding the noise to the top-5 highest-variance *(dashed red)* or lowest-variance components *(dashed blue)* per layer. While a model with perturbed high-variance components degrades the performance as the noise scale increases, a model with perturbed low-variance components consistently obtains high performance even with large perturbations. This shows that preserving the important weight components that respond to high-variance input components is critically important for effective neural network quantization.

## 3.2 CROSS-LAYER WEIGHT CORRELATION IMPACTS MODEL PERFORMANCE

So far, we have only described dependencies among weights within a single layer. However, dependencies between the weights across different layers also significantly impact the performance as well. To validate that, we perform layerwise sequential training from the bottom layer to the top. At the training of each layer, the model computes the principal components of the target layer and adds noise to its top-5 high/low components. As shown in Figure 3, progressive training with the low-variance components *(solid blue)* achieves significantly improved accuracy over the end-to-end training counterpart *(dashed blue)* with a high noise scale, which demonstrates the beneficial effect of modeling weight dependencies in earlier layers. We describe further details in the supplementary file.

## 4 TASK-DEPENDENT WEIGHT TRANSFORMATION FOR NN BINARIZATION

Our objective is to obtain binarized weights $w_q$ given pre-trained full-precision weights. We effectively mitigate performance degeneration from the binarization process by focusing on the inter-weight dependencies within each layer and across consecutive layers. Given a single-layered neural network parameterized by $w^{(1)}$, We first reformulate the quantization function $Q$ in Equation 2 with the weight correlation matrix $V^{(0)}$ and the importance vector $s^{(0)}$ so that each weight is disentangled from the others while allowing larger quantization errors on the unimportant disentangled weights (Unless otherwise stated, we omit the superscript denoting layer index):

$$Q(\boldsymbol{w}; \boldsymbol{s}, \boldsymbol{V}) = \underset{\boldsymbol{w}_q \in \mathbb{Q}}{\arg \min} \left\| \text{diag}(\boldsymbol{s}) \left( \boldsymbol{V}^\top \boldsymbol{w} - \boldsymbol{V}^\top \boldsymbol{w}_q \right) \right\|_F^2 + \gamma \left\| \boldsymbol{w}_q \right\|_1, \quad (6)$$

where $\boldsymbol{V} \in \mathbb{R}^{d_0 \times d_0}$, and $\boldsymbol{s} \in \mathbb{R}^{d_0}$ is a scaling term that assigns different importance scores to each row of $\boldsymbol{V}^\top \boldsymbol{w}$. We denote $\mathbb{Q} = \{\boldsymbol{\alpha} \odot \boldsymbol{b} : \boldsymbol{\alpha} \in \mathbb{R}^{d_1}, \boldsymbol{b} \in \{-1, 1\}^{d_0 \times d_1}\}$ as the set of possible binarized values for $\boldsymbol{w} \in \mathbb{R}^{d_0 \times d_1}$ with a scalar scaling factor for each output channel, where $\odot$ is an element-wise multiplication operator, with dimensions broadcasted appropriately. We additionally include $\ell_1$ norm adjusted by a hyperparameter $\gamma$. At the same time, we want our quantized model to minimize the empirical task loss (e.g., cross-entropy loss) for a given dataset. Thus we formulate the full objective in the form of a bilevel optimization problem to find the best quantized weights which minimize the task loss by considering the cross-layer weight dependencies and the relative importance among weights:

$$\boldsymbol{w}^*, \boldsymbol{s}^*, \boldsymbol{V}^* = \underset{\boldsymbol{w}, \boldsymbol{s}, \boldsymbol{V}}{\arg \min} \, \mathcal{L}_{task} \left( f\left( \mathcal{X}; \boldsymbol{w}_q \right), \mathcal{Y} \right), \quad \text{where } \boldsymbol{w}_q = Q(\boldsymbol{w}; \boldsymbol{s}, \boldsymbol{V}). \quad (7)$$

After the quantized training, the quantized weights $\boldsymbol{w}_q^*$ at layer $l$ are determined by $\boldsymbol{w}_q^* = Q(\boldsymbol{w}^*; \boldsymbol{s}^*, \boldsymbol{V}^*)$. In practice, directly solving the above bilevel optimization problem is impractical due to its excessive computational cost. We therefore consider the following relaxed problem:

$$\boldsymbol{\alpha}^*, \boldsymbol{w}^*, \boldsymbol{s}^*, \boldsymbol{V}^* = \underset{\boldsymbol{\alpha}, \boldsymbol{w}, \boldsymbol{s}, \boldsymbol{V}}{\arg \min} \, \mathcal{L}_{task} \left( f(\mathcal{X}; \boldsymbol{\alpha} \cdot \text{sgn}(\boldsymbol{w})), \mathcal{Y} \right) + \lambda \left\| \text{diag}(\boldsymbol{s}) \boldsymbol{V}^\top \left( \boldsymbol{w} - \boldsymbol{\alpha} \cdot \text{sgn}(\boldsymbol{w}) \right) \right\|_F^2$$
$$+ \gamma \left\| \boldsymbol{\alpha} \cdot \text{sgn}(\boldsymbol{w}) \right\|_1, \quad (8)$$

where $\lambda$ is a hyperparameter to balance between the quantization objective and task loss. Since it is impossible to compute the gradients for discrete values in quantized weights, we adopt the straight-through estimator Bengio et al. (2013) that is broadly used across QAT methods: $\text{sgn}(\boldsymbol{w})$ indicates the sign function applied elementwise to $\boldsymbol{w}$. We follow Liu et al. (2020) for the derivative of $\text{sgn}(\cdot)$. Finally, we obtain the desired quantized weights by $\boldsymbol{w}_q^* = \boldsymbol{\alpha} \cdot \text{sgn}(\boldsymbol{w}^*)$. In order to obtain the off-diagonal parts of the cross-layer dependency matrix $V$, we minimize Equation 8 with respect to $\boldsymbol{s}$ and $\boldsymbol{V}$ to dynamically determine the values (we omit $\mathcal{X}$ and $\mathcal{Y}$ from this argument for readability):

$$\mathcal{L}_{train}(\boldsymbol{\alpha}, \boldsymbol{w}, \boldsymbol{s}, \boldsymbol{V}) = \mathcal{L}_{task} \left( f\left( \mathcal{X}; \boldsymbol{\alpha} \cdot \text{sgn}(\boldsymbol{w}) \right), \mathcal{Y} \right) + \lambda \left\| \text{diag}(\boldsymbol{s}) \boldsymbol{V}^\top \left( \boldsymbol{w} - \boldsymbol{\alpha} \cdot \text{sgn}(\boldsymbol{w}) \right) \right\|_F^2$$
$$+ \gamma \left\| \boldsymbol{\alpha} \cdot \text{sgn}(\boldsymbol{w}) \right\|_1 + Reg(\boldsymbol{s}, \boldsymbol{V}), \quad (9)$$

where $Reg(\boldsymbol{s}, \boldsymbol{V}) := \|\boldsymbol{V} \boldsymbol{V}^\top - \boldsymbol{I}\|^2 + |\sigma - \sum_i \log(s_i)|^2$ is a regulariztion term which enforces $\boldsymbol{V}$ to be orthogonal and keeps the scale of $\boldsymbol{s}$ constant. Here, $\sigma$ is the constant initial value of $\sum_i \log(s_i)$, which is a non-negative importance score.

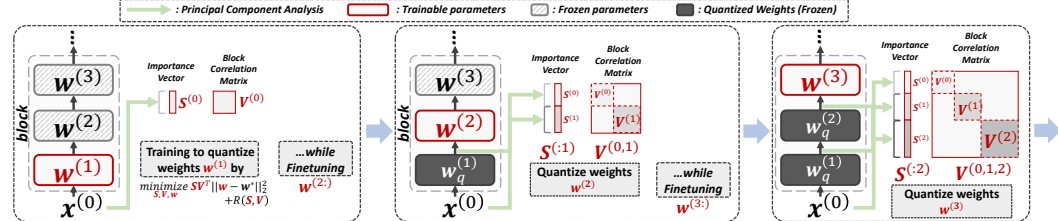

Figure 4: **Quantization-aware Training with BiTAT**: We perform a sequential training process: quantization training of a layer - rapid finetuning for upper layers. At each layerwise quantization, we also train the importance vector and orthonormal correlation matrix, which are initialized by PCA components of the current and lower layer inputs in the target block, and guide the quantization to consider the importance of disentangled weights.

## 4.1 Layer-progressive Quantization with Block-wise Weight Dependency

Now we extend our formulation for multi-layered neural networks considering cross-layer weight dependency. While we obtain the objective function in Equation 9, it is inefficient to perform quantization-aware training while considering the complete correlations of all weights in the given neural network. Therefore, we only consider cross-layer dependencies between only few consecutive layers (we denote it as a *block*), and initialize $s$ and $V$ using Principal Component Analysis (PCA) on the inputs to those layers within each block.

Formally, we define a weight correlation matrix in a neural network block $V^{(block)} \in \mathbb{R}^{(\sum_{i=1}^{k} d_i) \times (\sum_{i=1}^{k} d_i)}$, where $k$ is the number of layers in a block, similarly to the block-diagonal formulation in Li et al. (2021) to express the dependencies between weights across layers in the off-diagonal parts. We initialize $s^{(l)}$ and in-diagonal parts $V^{(l)}$ by applying PCA on the input covariance matrix:

$$V^{(block)} := \begin{bmatrix} V^{(1)} & & \cdots \\ & \ddots & \\ \cdots & & V^{(k)} \end{bmatrix}$$

Figure 5: Initialization of the block correlation matrix.

$$s^{(l)} \leftarrow (\boldsymbol{\lambda}^{(l)})^{\frac{1}{2}}, \quad V^{(l)} \leftarrow U^{(l)}, \qquad \text{where } U^{(l)} \boldsymbol{\lambda}^{(l)} (U^{(l)})^{\top} := \frac{1}{N} \sum_{i=1}^{N} o_i^{(l-1)} o_i^{(l-1)\top}, \quad (10)$$

where $o^{(l)}$ is a column vector and the output of $l$-th layer and $o^{(0)} = x$. This allows the weights at $l$-th layer to consider the dependencies on the weights from the earlier layers within the same neural block, and we refer to this method as *sequential quantization*, so that the model alleviates the quantization errors accumulated through propagating from the lower to higher consecutive layers while preserving the performance of the quantized model. Then, instead of having one set of $s$ and $V$ for each layer, we can keep the previous layer's $s$ and $V$ and expand them. Specifically, when quantizing layer $l$ which is a part of the block that starts with the layer $m$, we first apply PCA on the input covariance matrix to obtain $\boldsymbol{\lambda}^{(l)}$ and $U^{(l)}$. We then expand the existing $s^{(m:l-1)}$ and $V^{(m:l-1)}$ to obtain $s \in \mathbb{R}^{D+d_{l-1}}$ and $s \in \mathbb{R}^{D+d_{l-1}}$ as follows[*]:

$$[s^{(m:l)}]_i := \begin{cases} [s^{(m:l-1)}]_i, & i \leq D, \\ [(\boldsymbol{\lambda}^{(l)})^{\frac{1}{2}}]_{i-D}, & D < i, \end{cases} \qquad [V^{(m:l)}]_{i,j} := \begin{cases} [V^{(m:l-1)}]_{i,j}, & i,j \leq D, \\ [U^{(l)}]_{i-D,j-D}, & D < i,j, \\ 0, & \text{otherwise}, \end{cases} \quad (11)$$

where $D = \sum_{i=m}^{l-2} d_i$, as illustrated in Figure 4. The weight dependencies between different layers (i.e., off-diagonal areas) are trainable and zero-initialized. That is, at each layerwise quantization in the target block, we train the importance vector and orthonormal correlation matrix, where expanded areas are initialized by PCA components of the current layer inputs area. To enable the matrix multiplication of the weights with the expanded $s$ and $V$, we define the expanded block weights[†]:

$$w^{(m:l)} = \left[ \text{PadCol}(w^{(m:l-1)}, d_l); w^{(l)} \right], \quad (12)$$

where $\text{PadCol}(\cdot, c)$ zero-pads the input matrix to the right by $c$ columns. Then, our final objective from Equation 9 with cross-layer dependencies is given as follows:

---

[*] $[\cdot]_i$ indicates the $i$-th element of the object inside the brackets.

[†] $[A; B]$ indicates vertical concatenation of the matrices A and B.

---

**Algorithm 1** Neural Network Binarization with Task-dependent Aggregated Transformation

---

1: **Input:** Pre-trained weights $\boldsymbol{w}^{(1)}, \dots, \boldsymbol{w}^{(L)}$ for $L$ layers, task loss function $\mathcal{L}$, Maximum size of input-dimension group $k$, quantization epochs per layer $N_{ep}$.
2: **Output:** Quantized weights $\boldsymbol{w}^{*(1)}, \dots, \boldsymbol{w}^{*(L)}$.
3: $\mathcal{B}_1, \dots, \mathcal{B}_n \leftarrow$ Divide the neural network into $n$ blocks
4: **for each** block $\mathcal{B}$ **do**
5:     $\boldsymbol{s} = [\,], \boldsymbol{V} = [\,]$
6:     **for each** layer $l$ in $\mathcal{B}$ **do**
7:         $\boldsymbol{o}^{(l-1)} \leftarrow$ inputs for layer $l$
8:         $\boldsymbol{P} \leftarrow$ **if** $d_{l-1} > k$ **then** K-MEANS$(\boldsymbol{X}^{(l)}, k)$ **else** $\boldsymbol{I}_{d_{l-1}}$           $\triangleright$ Grouping permutation matrix
9:         $\boldsymbol{U}\,\mathrm{diag}(\boldsymbol{\lambda})\boldsymbol{U}^{\top} = $ PCA$(\boldsymbol{P}\boldsymbol{o}^{(l-1)})$           $\triangleright$ Initialization values
                                                       $\triangleright$ for the expanded part
10:         $\boldsymbol{s} \leftarrow [\boldsymbol{s}; \boldsymbol{\lambda}^{\frac{1}{2}}], \boldsymbol{V} \leftarrow \begin{bmatrix} \boldsymbol{V} & \boldsymbol{0} \\ \boldsymbol{0} & \boldsymbol{U} \end{bmatrix}$           $\triangleright$ expand $s$ and $V$
11:         $\mathbf{s}\alpha^{(l:L)}, \boldsymbol{w}^{(l:L)}, \boldsymbol{s}, \boldsymbol{V} \leftarrow \arg\min_{\mathbf{s}\alpha, \boldsymbol{w}, \boldsymbol{s}, \boldsymbol{V}} \mathcal{L}_{train}(\mathbf{s}\alpha, \boldsymbol{w}, \boldsymbol{s}, \boldsymbol{V})$   $\triangleright$ Iterate for $N_{ep}$ epochs
12:         $\boldsymbol{w}_q^{(l)} \leftarrow \mathbf{s}\alpha^{(l)} \cdot \mathrm{sgn}(\boldsymbol{w}^{(l)})$

---

$$\mathcal{L}_{train}(\boldsymbol{w}^{(l:L)}, \boldsymbol{s}^{(m:l)}, \boldsymbol{V}^{(m:l)}) = \mathcal{L}_{task}\left(f\left(\mathcal{X}; \{\alpha \cdot \mathrm{sgn}(\boldsymbol{w}^{(l)}), \boldsymbol{w}^{(l+1:L)}\}\right), \mathcal{Y}\right)$$

$$+ \lambda \left\| \mathrm{diag}(\boldsymbol{s}^{(m:l)}){\boldsymbol{V}^{(m:l)}}^{\top}\left(\boldsymbol{w}^{(m:l)} - \alpha \cdot \mathrm{sgn}(\boldsymbol{w}^{(m:l)})\right)\right\|_F^2 \quad (13)$$

$$+ \gamma \left\| \alpha \cdot \mathrm{sgn}(\boldsymbol{w}^{(m:l)})\right\|_1 + Reg(\boldsymbol{s}^{(m:l)}, \boldsymbol{V}^{(m:l)}).$$

Given the backbone architecture with $L$ layers, we minimize $\mathcal{L}_{train}(\boldsymbol{w}^{(l)}, \boldsymbol{s}^{(l)}, \boldsymbol{V}^{(l)})$ with respect to $\boldsymbol{w}^{(l)}, \boldsymbol{s}^{(l)}$, and $\boldsymbol{V}^{(l)}$ to find the desired binarized weights $\boldsymbol{w}_q^{*(l)}$ for layer $l$ while keeping the other layers frozen. Next, we finetune the following layers using the task loss function a few epochs before performing QAT on following layers, as illustrated in Figure 4. This sequential quantization proceeds from the bottom layer to the top and the obtained binarized weights are frozen during the training.

## 4.2 COST-EFFICIENT BiTAT VIA AGGREGATED WEIGHT CORRELATION USING REDUCTION MATRIX

We derived a QAT formulation which focues on the cross-layer weight dependency by learning block-wise weight correlation matrices. Yet, as the number of inputs to higher layers is often large, the model constructs higher-dimensional $\boldsymbol{V}^{(l)}$ on upper blocks, which is costly. In order to reduce the training memory footprint as well as the computational complexity, we aggregate the input dimensions into several small groups based on functional similarity using $k$-means clustering.

First, we take feature vectors, the outputs of the $l$-th layer $\boldsymbol{o}_1^{(l)}, \dots, \boldsymbol{o}_N^{(l)} \in \mathbb{R}^{d_l}$ for each output dimension, to obtain $d_l$ points $\boldsymbol{p}_1, \boldsymbol{p}_2, \dots, \boldsymbol{p}_{d_l} \in \mathbb{R}^N$, then aim to cluster the points to $k$ groups using $k$-means clustering, each containing $N/k$ points. Let $g_i \in \{1, 2, \dots, k\}$ indicate the group index of $\boldsymbol{p}_i$, for $i = 1, \dots, d_l$. We construct the reduction matrix $\boldsymbol{P} \in \mathbb{R}^{k \times d_l}$, where $P_{ij} = \frac{1}{N/k}$ if $g_j = i$, and 0 otherwise. Each group corresponds to a single row of the reduced $\widehat{\boldsymbol{V}}^{(l+1)} \in \mathbb{R}^{k \times k}$ instead of the original dimension $d_l \times d_l$. In practice, this **significantly reduces the memory consumption** of the $\boldsymbol{V}$ (down to **0.07%**). Now, we replace $\boldsymbol{s}$ and $\boldsymbol{V}^{\top}$ in Equation 13 to $\widehat{\boldsymbol{s}}$ and $\widehat{\boldsymbol{V}}^{\top}\boldsymbol{P}$, respectively, initializing $\widehat{\boldsymbol{s}}$ and $\widehat{\boldsymbol{V}}$ with the grouped input covariance $\frac{1}{N}\sum_{i=1}^{N}(\boldsymbol{P}\boldsymbol{o}_i^{(l)})(\boldsymbol{P}\boldsymbol{o}_i^{(l)})^{\top}$. We describe the full training process of our proposed method in Algorithm 1. The total number of training epochs taken in training is $O(LN_{ep})$, where $L$ is the number of layers, and $N_{ep}$ is the number of epochs for the quantizing step for each layer.

## 5 EXPERIMENTS

We validate a new quantization-aware training method, BiTAT, over multiple benchmark datasets; CIFAR-10, CIFAR-100 Krizhevsky et al. (2009), and ILSVRC2012 ImageNet Deng et al. (2009) datasets. We use MobileNet V1 (Howard et al., 2017) backbone network, which is a compact neural architecture designed for mobile devices. We follow overall experimental setups from prior works Yamamoto (2021); Liu et al. (2020).

Table 1: **Performance comparison of BiTAT with baselines.** We report the averaged test accuracy across three independent runs. The best results are highlighted in bold, and results of cost-expensive models ($10^8 \uparrow$ ImgNet FLOPs) are de-emphasized in gray. We refer to several results reported from their own papers, denoted as $^{\dagger}$.

| METHODS | ARCHITECTURE | BITWIDTH WEIGHT / ACTIV. | IMGNET FLOPs ($\times 10^7$) | IMGNET ACC (%) | CIFAR-10 ACC (%) | CIFAR-100 ACC (%) |
|---|---|---|---|---|---|---|
| Full-precision | ResNet-18 | 32 / 32 | 200.0 | 69.8 | 93.02 | 75.61 |
| | MobileNet V1 | 32 / 32 | 56.90 | 70.6 | - | 66.68 |
| | MobileNet V2 | 32 / 32 | 31.40 | 71.9 | 94.43 | 68.08 |
| BRECQ Li et al. (2021) | MobileNet V2 | 4 / 4 | 3.31 | 66.57$^{\dagger}$ | - | - |
| DBQ Dbouk et al. (2020) | MobileNet V2 | 4 / 8 | 3.60 | 70.54$^{\dagger}$ | 93.77 | 73.20 |
| LCQ Yamamoto (2021) | ResNet-18 | 2 / 2 | 15.00 | 68.9$^{\dagger}$ | - | - |
| | MobileNet V2 | 4 / 4 | 3.31 | 70.8$^{\dagger}$ | - | - |
| MeliusNet59 Bethge et al. (2020) | N/A | 1 / 1 | 24.50 | 70.7$^{\dagger}$ | - | - |
| Bi-Real Net Liu et al. (2018) | ResNet-18 | 1 / 1 | 15.00 | 56.4$^{\dagger}$ | - | - |
| Real-to-Bin Martinez et al. (2020) | ResNet-18 | 1 / 1 | 15.00 | 65.4$^{\dagger}$ | - | 76.2$^{\dagger}$ |
| EBConv Bulat et al. (2021) | ResNet-18 | 1 / 1 | 11.00 | 71.2$^{\dagger}$ | - | 76.5$^{\dagger}$ |
| ReActNet-C Liu et al. (2020) | MobileNet V1 | 1 / 1 | 14.00 | 71.4$^{\dagger}$ | 90.77 | 67.41 |
| ReActNet-A Liu et al. (2020) | MobileNet V1 | 1 / 1 | **1.20** | 68.26 | 89.73 | 65.51 |
| **BiTAT-C (Ours)** | MobileNet V1 | 1 / 1 | 14.00 | - | - | **69.45** |
| **BiTAT-A (Ours)** | MobileNet V1 | 1 / 1 | **1.20** | **68.51** | **90.21** | **68.36** |

**Baselines and training details.** While our method aims to solve the QAT problem, we extensively compare our *BiTAT* against various methods; Post-training Quantization (PTQ) method: BRECQ Li et al. (2021), and Quantization-aware Training (QAT) methods: DBQ Dbouk et al. (2020), EBConv Bulat et al. (2021), Bi-Real Net Liu et al. (2018), Real-to-Bin Martinez et al. (2020), LCQ Yamamoto (2021), MeliusNet Bethge et al. (2020), ReActNet Liu et al. (2020). Note that DBQ, LCQ, and MeliusNet, keep some crucial layers, such as $1 \times 1$ downsampling layers, in full-precision, leading to inefficiency at evaluation time. Due to the page limit, we provide the details on baselines and the training and inference phase during QAT including hyperparameter setups in the Supplementary file.

## 5.1 QUANTITATIVE ANALYSIS

We compare our BiTAT against various PTQ and QAT-based methods in Table 1 on multiple datasets. BRECQ introduces an adaptive PTQ method by focusing on the weight dependency via hessian matrix computations, resulting in significant performance deterioration and excessive training time. DBQ and LCQ suggest QAT methods, but the degree of bitwidth compression for the weights and activations is limited to 2- to 8-bits, which is insufficient to meet our interest in achieving neural network binarization with 1-bits weights and activations. MeliusNet only suffers a small accuracy drop, but it has a high OP count. DBQ and LCQ restrict the bit-width compression to be higher at 4 bits so that they cannot enjoy the XNOR-Bitcount optimization for speedup. Although Bi-Real Net, Real-to-Bin, and EBConv successfully achieve neural network binarization, over-parameterized ResNet is adopted as backbone networks, resulting in higher OP count. Moreover, except EBConv, these works still suffer from a significant accuracy drop. ReActNet binarizes all of the weights and activations (except the first and last layer) in compact network architectures while preventing model convergence failure. Nevertheless, the method still suffers from considerable performance degeneration of the binarized model. On the other hand, our BiTAT prevents information loss during quantized training up to 1-bits, showing a superior performance than ReActNet, 0.37 % $\uparrow$ for ImageNet, 0.53% $\uparrow$ for CIFAR-10, and 2.31% $\uparrow$ for CIFAR-100. Note that BiTAT further achieves on par performance of the MobileNet backbone for CIFAR-100. The results support our claim on layer-wise quantization from the bottom layer to the top, reflecting the disentangled weight importance and correlation with the quantized weights at earlier layers.

**Ablation study** We conduct ablation studies to analyze the effect of salient components in our proposed method in Figure 7 Left. BiTAT based on layer-wise sequential quantization without weight transformation already surpasses the performance of ReActNet, demonstrating that layer-wise progressive QAT through an implicit reflection of adjusted importance plays a critical role in preserving the pre-trained models during quantization. We adopt intra-layer weight transformation using the input-dependent orthonormal matrix, but no significant benefits are observed. Thus, we expect that only disentangling intra-layer weight dependency is insufficient to fully reflect the adjusted importance of each weight due to a binarization of earlier weights/activations. This is evident that BiTAT considering both intra-layer and cross-layer weight dependencies achieves improved performance than the case with only intra-layer dependency. Yet, this requires considerable additional training time to compute

| METHOD | INTRA-LAYER TRANSFORM | CROSS-LAYER TRANSFORM | Accuracy (%) | Train Time (hours) |
|---|---|---|---|---|
| REACTNET LIU ET AL. (2020) | N/A | N/A | $65.51 \pm 0.74$ | 10.75 |
| BiTAT (Ours) | $\times$ | $\times$ | $68.17 \pm 0.07$ | 3.49 |
| | $\checkmark$ | $\times$ | $67.82 \pm 0.22$ | 3.66 |
| | $\checkmark$ | $\checkmark$ | $68.21 \pm 0.24$ | 8.50 |
| *w/ Filter-wise Transform* | | | $67.86 \pm 0.11$ | 3.01 |
| *w/ Truncated SVD* | | | $67.55 \pm 1.04$ | 3.44 |
| ***w/ Aggregated Transform*** | | | **$68.36 \pm 0.45$** | **3.11** |

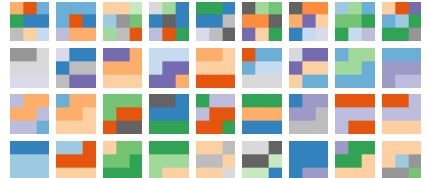

Figure 7: **Left: Ablation study** for analyzing core components in our method. We report the averaged performance and 95% confidence interval across 3 independent runs and the complete BiTAT result is highlighted in gray background. **Right: Visualization of the weight grouping** during sequential quantization of BiTAT. Each 3×3 square represents a convolutional filter of the topmost layer ($26^{th}$, excluding the classifier) of our model, and each unique color represents each group to which weight elements belong.

with a chunk-wise transformation matrix. In the end, BiTAT with aggregated transformations, which is our full method, outperforms our defective variants in both terms of model performance and training time by drastically removing redundant correlation through reduction matrices. We note that using $k$-means clustering for aggregated correlation is also essential, as another variant, BiTAT with filter-wise transformations, which filter-wisely aggregates the weights instead, results in deteriorated performance.

## 5.2 QUALITATIVE ANALYSIS

**Visualization of Reduction Matrix** We visualize the weight grouping for BiTAT in Figure 7 Right

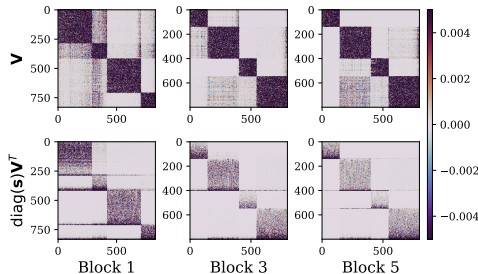

Figure 6: Visualization of the learned $V$ matrix and the $\mathrm{diag}(s)V^{\top}$ of three blocks of the network, with the CIFAR-100 dataset. Notice the off-diagonal parts which represent cross-layer dependencies.

to analyze the effect of the reduction matrix, which groups the weight dependencies in each layer based on the similarity between the input dimensions. Each 3×3 square represents a convolutional filter, and each unique color in weight elements represents which group each weight is assigned to, determined by the $k$-means algorithm, as described in Section 4.2. We observe that weight elements in the same filter do not share their dependencies; rather, on average, they often belong to four-five different weight groups. Opposite to these observations, BRECQ regards the weights in each filter as the same group for computing the dependencies in different layers, which is problematic since weight elements in the same filter can behave differently from each other.

**Visualization of Cross-layer Weight Dependency** In Figure 6, we visualize learned transformation matrices $V$ (*top row*), which shows that many weight elements at each layer are also dependent on other layer weights as highlighted in darker colors, verifying our initial claim. Further, we provide visualizations for their multiplications with corresponding importance vectors $\mathrm{diag}(s)V^{\top}$ (*bottom row*). Here, the row of $V^{\top}$ is sorted by the relative importance in increasing order at each layer. We observe that important weights in a layer affect other layers, demonstrating that cross-layer weight dependency impacts the model performance during quantized training.

## 6 CONCLUSION

In this work, we explored long-overlooked factors that are crucial in preventing the performance degeneration with extreme neural network quantization: the inter-weight dependencies. That is, quantization of a set of weights affect the weights for other neurons within each layer, as well as weights in consecutive layers. Grounded by the empirical analyses of the node interdependency, we propose a Quantization-Aware Training (QAT) method for binarizing the weights and activations of a given neural network with minimal loss of performance. Specifically, we proposed orthonormal transformation of the weights at each layer to disentangle the correlation among the weights to minimize the negative impact of quantization on other weights. Further, we learned scaling term to allow varying degree of quantization error for each weight based on their measured importance, for layer-wise quantization. Then we proposed an iterative algorithm to perform the layerwise quantization in a progressive manner. We demonstrate the effectiveness of our method in neural network binarization on multiple benchmark datasets with compact backbone networks, largely outperforming state-of-the-art baselines.

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
