# OpenReview forum: "BiTAT: Neural Network Binarization with Task-Dependent Aggregated Transformation"
_ICLR.cc/2023/Conference — Submitted to ICLR 2023_

### Official Review · Reviewer_axSm · 2022-10-21

**Confidence:** 4
**Clarity, Quality, Novelty And Reproducibility:** The clarity of the paper can be furth…
**Correctness:** 2
**Technical Novelty And Significance:** 3
**Empirical Novelty And Significance:** 3
**Recommendation:** 3

**Strength And Weaknesses:**

Strength:
The work gives some improvement over ReActNet on MobileNet V1-like architecture.

Weakness:
The authors only show improvement in single-model architecture, and the difference is relatively small.
It is not clear which component is responsible for the improvement, and it is also not clear whether pre-training is necessary.

Actionable feedback:
1. Additional experiments on other model architectures.
2. The difference with and without aggregated transformation training is relatively small, given the variance of the experiment results. More experiments are also needed here.
3. An ablation experiment on a model without pre-training would make the paper stronger.
4. Please fix the citation format of the appendix.

**Summary Of The Paper:**

This work proposes to improve binary neural network performance through layer-wise quantization and orthonormal transformation of weights.

**Summary Of The Review:**

The paper is not ready for publication, and additional experiments are needed to identify the reason behind the improvement and make their claim solid.

---

> ### Author Response · Authors · 2022-11-18
> **Response**
>
> **(1) The difference with and without aggregated transformation training is relatively small, given the variance of the experiment results. More experiments are also needed here.**
>
> The point of aggregated transformation is to reduce the computational cost of the training, not to improve the accuracy. In Figure 7, compare the model in the 3rd row of BiTAT (version without aggregated transform) with the final row of BiTAT (with aggregated transform). We can observe that without aggregated transform, the training process is much longer, but with aggregated transform, thanks to the reduced cost of computing the $V$ matrices, we have a much shorter training time.
>
> **(2) An ablation experiment on a model without pre-training would make the paper stronger.**
>
> Pre-training is utilized in all QAT methods including ReActNet. This is because regardless of the which QAT method is used, without pre-training, the model sufferes a large drop in accuracy, because QAT assumes that we start from a fully-trained full-precision model.
>
> **(3) Please fix the citation format of the appendix.**
>
> Thank you for pointing out the citation format in the appendix. We will fix it in the revision.

---

### Official Review · Reviewer_2eTK · 2022-10-24

**Confidence:** 5
**Correctness:** 3
**Technical Novelty And Significance:** 3
**Empirical Novelty And Significance:** Not applicable
**Recommendation:** 5

**Clarity, Quality, Novelty And Reproducibility:**

The paper has been well structured, and some of the ideas seem to be novel. Reproducibility appears to fair.

**Strength And Weaknesses:**

Strengths:

- The paper is well written and easy to follow.
- The proposed idea of principle components initialization is novel.
- The experiments are well conducted.

Weaknesses:

- The experimental results are not convincing enough. One of the claimed contributions is the enhancement of intra-layer and cross-layer dependencies through the learnable correlation weight matrix. But, we can see in figure 7 left, there is only marginal improvements compare to the baseline, i.e., imagenet top-1 68.17% → 68.36%.
Second, the reported result based on ReActNet-A (in table 1) is about  0.89% lower (68.26% → 69.4%) than that reported in the ReActNet paper,  which is inconsistent with the claim “… it significantly outperforms baselines with the compact neural network architecture. “ Considering that there is rarely reproducibility problems reported with the ReActNet’s codes (training scripts, hyper-parameters, pre-trained models all available.), I would recommend that the author be able to give a suitable explanation.

- I was wondering how much additional overhead the layer-wise progressive QAT will create compared to the classical baseline methods and combined with the marginal accuracy gain, is the extra overhead worth it?

- It will be very interesting to see the performance on a second backbone, e.g., BiRealNet-18. The official ReActNet code actually also supports it, and this model is smaller, thus easier to train.

- I recommend establishing a mathematical justification for the weight projection using the basis of the activation values. I am not sure for this to be convincing enough.


**Summary Of The Paper:**

This paper proposes to consider the important correlations between weights and activations in each binary NN layer. Second, the authors train a correlation matrix to optimize the cross-layer weight correlations. They further propose to apply QAT to learn the correlations and progressively quantize the weights from sequential layers rather than quantizing the weights from each layer independently.

**Summary Of The Review:**

Though the paper is well written, there are likewise specific novel ideas. However, hopefully, the author can answer a few concerns I raised, and these concerns make the paper borderline.

---

> ### Author Response · Authors · 2022-11-18
> **Response**
>
> **(1) The experimental results are not convincing enough. One of the claimed contributions is the enhancement of intra-layer and cross-layer dependencies through the learnable correlation weight matrix. But, we can see in figure 7 left, there is only marginal improvements compare to the baseline, i.e., imagenet top-1 68.17\% → 68.36\%.**
>
> The increase in performance from 68.17\%($\pm$0.07) to 68.36\%($\pm0.45$) is hardly marginal. Additionally, note that the ``baseline'' with the accuracy 68.17\% already contains the layer-progressive quantization method, which is another one of our contributions as well.
>
> **(2) the reported result based on ReActNet-A (in table 1) is about 0.89\% lower (68.26\% → 69.4\%) than that reported in the ReActNet paper, which is inconsistent with the claim “… it significantly outperforms baselines with the compact neural network architecture. “ Considering that there is rarely reproducibility problems reported with the ReActNet’s codes (training scripts, hyper-parameters, pre-trained models all available.), I would recommend that the author be able to give a suitable explanation.**
>
> We used the implementation in https://github.com/liuzechun/ReActNet to evaluate ReActNet-A, and obtained the results stated in our paper.
>
> **(3) I was wondering how much additional overhead the layer-wise progressive QAT will create compared to the classical baseline methods and combined with the marginal accuracy gain, is the extra overhead worth it?**
>
> In Figure 7 Left, we compare the training time of ReActNet, a classic QAT method, with our method. We observe that our method can achieve better results with much shorter GPU training time.
>
> **(4) I recommend establishing a mathematical justification for the weight projection using the basis of the activation values. I am not sure for this to be convincing enough.**
>
> We elaborated on this in the beginning of Section 3, as well as in Section 3.1. To reiterate, given a dataset, if the input vectors to a linear transformation layer (such as convolutional or matrix multiplication) has correlated elements, we can use that as an advantage in order to find a good-performing quantized weights more efficiently.

---

### Official Review · Reviewer_3kJB · 2022-10-24

**Confidence:** 4
**Correctness:** 3
**Technical Novelty And Significance:** 3
**Empirical Novelty And Significance:** 2
**Recommendation:** 5

**Clarity, Quality, Novelty And Reproducibility:**

This paper has an interesting formulation for target-dependent importance measures for QAT. The idea is straightforward, and the realization of the idea in practice is appealing. Moreover, because the source code is provided, I convince of the reproducibility of the idea.

**Strength And Weaknesses:**

The relationship between weight/activation quantization and the information within generated feature map has been explored in several previous studies, but this paper proposes an intuitive formulation of task-dependent weight correlation. The writing is well organized, so the complex part of the idea becomes straightforward via the development of paragraphs.

However, this paper has a few limitations that adversely affect the final evaluations. First, while the paper mainly focuses on the binary neural network, isn't it possible to extend the proposed method for multi-bit quantization? While binary quantization has a lot of advantages, multi-bit precision has substantial practical importance; thereby, presenting the corresponding results would be better. Second, BiTAT-A networks show outstanding accuracy in the CIFAR-100 dataset, but the gain becomes negligible in the ImageNet dataset. I highly doubt that the benefit of the proposed method is shrunk in a large, complex dataset. Can we still exploit the benefit of the proposed method in a large network for a complex dataset? Supportive material is required. Last, can we apply the proposed method for other complicated vision tasks, e.g., object detection or super-resolution?

**Summary Of The Paper:**

This paper proposes a novel quantization-aware training scheme for the binary neural network. The novelty of this paper is the formulation of error minimization during quantization considering inter-weight dependencies between the weights within each layer and across consecutive layers. Based on the principle component analysis for activation, we can estimate the importance of feature characteristics, and QAT should optimize the quantization error to minimize the critical error of the generated feature after the quantization. The task-dependent weight transformation is well formularized, and the layer-wise progressive quantization is proposed on top of the transformation, resulting in high accuracy after QAT.

**Summary Of The Review:**

Personally, I like the idea of this paper, and the quality of the paper is quite good. However, the idea seems not yet validated enough. I can't ensure we are able to apply the idea beyond the toy-level scale problems. It would be better to provide additional extensive studies.

---

> ### Author Response · Authors · 2022-11-18
> **Response**
>
> **(1) First, while the paper mainly focuses on the binary neural network, isn't it possible to extend the proposed method for multi-bit quantization? While binary quantization has a lot of advantages, multi-bit precision has substantial practical importance; thereby, presenting the corresponding results would be better.**
>
> It is possible by replacing the $\alpha \cdot \mathrm{sgn}(w)$ terms in our equation with a limited-precision rounding function. However, multi-bit quantization is inferior to binarization in terms of computational cost in inference and storage cost. Since we can already achieve near-full-precision accuracies in a compact architecture such as MobileNet, we saw no point in demonstrating BiTAT in multi-bit quantization scenarios.
>
> **(2) BiTAT-A networks show outstanding accuracy in the CIFAR-100 dataset, but the gain becomes negligible in the ImageNet dataset. I highly doubt that the benefit of the proposed method is shrunk in a large, complex dataset. Can we still exploit the benefit of the proposed method in a large network for a complex dataset? Supportive material is required.**
>
> ImageNet is an extremely large dataset with high information content compared to CIFAR-100. Therefore, it is natural to expect a bigger drop in test accuracy, since the binarized model space is inherently less expressive than the full-precision model space. However, the test accuracy drop is still only about 2\% compared to the full-precision model, and we still achieve better accuracy than ReActNet.
>
> **(3) Can we apply the proposed method for other complicated vision tasks, e.g., object detection or super-resolution?**
>
> Our methdology only requires that the deep learning models use some kind of linear operations in its internal layers, such as convolutions or matrix multiplications. Therefore, BiTAT can be applied to any kind of tasks, not limited to vision.

---

### Official Review · Reviewer_Ed2K · 2022-10-25

**Confidence:** 4
**Clarity, Quality, Novelty And Reproducibility:** See Strength And Weaknesses. The code…
**Correctness:** 2
**Technical Novelty And Significance:** 2
**Empirical Novelty And Significance:** 2
**Recommendation:** 3

**Details Of Ethics Concerns:**

The methodology and visualization seems to be exhaustively similar to the previous BRECQ [1], albeit the two in a different orientation (1-bit QAT vs. 2~8-bit PTQ).
[1] Yuhang Li, et al. BRECQ: Pushing the limit of post-training quantization by block reconstruction. ICLR, 2021.

**Strength And Weaknesses:**

The strengths are as follows:
(1) Benefiting from the proposed method, the performance of BNN is improved to a certain extent, including on ImageNet and CIFAR100 datasets;
(2) The paper is well-written and visualized.

But there are some significant weaknesses:
(1) My biggest concern is that this paper can almost be seen as a direct application of the techniques and ideas from the PTQ method BRECQ to binarized perception training, which not only limits the novelty of the method but may even be motivated wrongly.
Since in the PTQ task, the weights of the model need to be quantized without retraining, methods such as BRECQ (such as AdaRound, QDrop, etc.) are devoted to evaluating the impact of quantization on the model. where BRECQ takes into account the intra-block weight dependency and reconstructs the quantized model block-wise. With QAT, however, a fundamental difference is that the weights are aware of the influence of the quantizer and optimized. An empirical conclusion is that the lower the quantization bit width, the less similar the weights of the trained quantized model are to the original model. Some 1-bit binarization work even pointed out that panorama pre-weights after retraining in binary neural networks should be optimized to a bimodal distribution, completely independent of full accuracy.
Therefore, the paper first needs to prove theoretically or empirically that the dependency of weights in 1-bit QAT still needs to be maintained in the context of being highly discretized, rather than being optimized to obtain new dependencies. Moreover, in binarization-aware quantization, the back-propagation-based optimization of weights actually takes into account the effect of each weight quantization on the global prediction (although discretization causes gradient errors). On this basis, why limit the dependency of weights?
(2) Moreover, compared with many recent works, the modified work is far from realizing SoTA, such as [1][2][3][4]. The accuracy of ReActNet-A and the proposed BiTAT-A is even far lower than what was reported in the original paper [1]. Also, it would be unwise to compare QAT with PTQ as a comparison method unless the QAT method has a tiny training computational cost comparable to PTQ.

[1] Zechun Liu, et al. ReActNet: Towards precise binary neural network with generalized activation functions. ECCV, 2020.
[2] Zechun Liu, et al. How do adam and training strategies help bnns optimization. ICML, 2021.
[3] Zhijun Tu, et al. Adabin: Improving binary neural networks with adaptive binary sets. arXiv, 2022.
[4] Sheng Xu, et al. Recurrent bilinear optimization for binary neural networks. arXiv, 2022.

**Summary Of The Paper:**

This paper proposes a QAT method to alleviate performance degeneration with binarization by focusing on the inter-weight dependencies, between the weights within each layer and across consecutive layers. To minimize the quantization impact of each weight on others, the authors perform an orthonormal transformation of the weights at each layer by training an input-dependent correlation matrix and importance vector, such that each weight is disentangled from the others. Then, the authors quantize the weights based on their importance to minimize the loss of information from the original weights/activations. The authors further perform progressive layer-wise quantization from the bottom layer to the top, so that quantization at each layer reflects the quantized distributions of weights and activations at previous layers.

**Summary Of The Review:**

See Strength And Weaknesses.

---

> ### Author Response · Authors · 2022-11-18
> **Response**
>
> **(1) My biggest concern is that this paper can almost be seen as a direct application of the techniques and ideas from the PTQ method BRECQ to binarized perception training, which not only limits the novelty of the method but may even be motivated wrongly.**
>
> Our method is not a direct application of BRECQ, rather, it is motivated from a different viewpoint that ended up to be superficially a bit similar to BRECQ in its methodology. Our motivation comes from observing applying PCA to the input of the layers, whereas BRECQ's motivation comes from the 2nd order approximation of the quantization error.
>
> **(2) The paper first needs to prove theoretically or empirically that the dependency of weights in 1-bit QAT still needs to be maintained in the context of being highly discretized, rather than being optimized to obtain new dependencies. Moreover, in binarization-aware quantization, the back-propagation-based optimization of weights actually takes into account the effect of each weight quantization on the global prediction (although discretization causes gradient errors). On this basis, why limit the dependency of weights?**
>
> We suspect that you might have a misunderstanding about the term "dependency" in our paper. In our paper, "dependency" refers to the property of the weight elements in a NN model acting together. Moreover, we update and obtain new "dependencies" between the weight elements at every training step. Please see Section 3 on the top of Page 4 for the in-depth explanation and motivation.
>
> **(3) Moreover, compared with many recent works, the modified work is far from realizing SoTA, such as [1][2][3][4]. The accuracy of ReActNet-A and the proposed BiTAT-A is even far lower than what was reported in the original paper [1]**
>
> We used the implementation in https://github.com/liuzechun/ReActNet to evaluate ReActNet-A, and obtained the results stated in our paper.
>
> **(4) It would be unwise to compare QAT with PTQ as a comparison method unless the QAT method has a tiny training computational cost comparable to PTQ.**
>
> The reason for including PTQ methods in our results table is to demonstrate the limitation of PTQ in a setting where we have an abundant training computational cost budget. We also wanted to highlight the advantage of our BiTAT over a (superficially) similar method, BRECQ, in that scenario.

---

### Decision · Program_Chairs · 2023-01-20

**Decision:**

Reject

**Justification For Why Not Higher Score:**

All reviewers lean toward reject.  The AC believes the author response did not sufficiently address reviewer concerns about experimental weaknesses.

**Justification For Why Not Lower Score:**

N/A

**Metareview: Summary, Strengths And Weaknesses:**

This paper proposes to improve binarization of neural networks by accounting for inter-weight dependencies in a scheme (BiTAT) that quantizes layers in sequential order.  Some concerns are each raised by multiple reviewers, including novelty of the approach and whether the experimental results are sufficiently convincing.  All reviewers lean toward rejecting the paper.

The author response addresses some reviewer questions, but does not present any additional results that might address the experimental concerns.  Specifically, Reviewer 3kJB points out that BiTAT has negligible gain on ImageNet with respect to ReActNet-A [Liu et al., 2020].  Reviewer 2eTK points out that ablation experiments in Figure 7 show only marginal improvement over the baseline due to the proposed techniques.  While authors argue otherwise, the Area Chair agrees with the reviewer characterizations of these improvements as marginal.

Overall, the Area Chair agrees with reviewers that the experimental case for the proposed method is insufficiently convincing, and as a result cannot recommend accepting the paper.